# Oregano (*Origanum vulgare*) Consumption Reduces Oxidative Stress and Markers of Muscle Damage after Combat Readiness Tests in Soldiers

**DOI:** 10.3390/nu15010137

**Published:** 2022-12-28

**Authors:** Hossein Shirvani, Behzad Bazgir, Alireza Shamsoddini, Ayoub Saeidi, Seyed Morteza Tayebi, Kurt A. Escobar, Ismail Laher, Trisha A. VanDusseldorp, Katja Weiss, Beat Knechtle, Hassane Zouhal

**Affiliations:** 1Exercise Physiology Research Center, Life Style Institute, Baqiyatallah University of Medical Sciences, Tehran 1435916471, Iran; 2Department of Physical Education and Sport Sciences, Faculty of Humanities and Social Sciences, University of Kurdistan, Kurdistan, Sanandaj 6617715175, Iran; 3Department of Exercise Physiology, Faculty of Physical Education and Sport Science, Allameh Tabataba’i University, Tehran 1489684511, Iran; 4Department of Kinesiology, California State University, Long Beach, CA 90840, USA; 5Department of Anesthesiology, Pharmacology and Therapeutics, The University of British Columbia, Vancouver, BC V6T 1Z3, Canada; 6Bonafide Health, LLC p/b JDS Therapeutics, 500 Mamaroneck Ave, Harrison, NY 10528, USA; 7Institute of Primary Care, University of Zurich, 8091 Zurich, Switzerland; 8Medbase St. Gallen Am Vadianplatz, 9000 St. Gallen, Switzerland; 9Laboratoire Mouvement, Sport, Santé, University of Rennes, M2S—EA 1274, 35000 Rennes, France; 10Institut International des Sciences du Sport (2I2S), 35850 Irodouer, France

**Keywords:** military exercise, antioxidant capacity, muscular indices, oxidative stress, soldiers

## Abstract

Military activities often involve high-intensity exercise that can disrupt antioxidant capacity. We investigated the effects of oregano supplementation on muscle damage, oxidative stress, and plasma antioxidant markers of soldiers performing the army combat readiness test (ACRT). Twenty-four healthy male soldiers (age: 24 ± 3 years, height: 167 ± 14 cm, mass: 66 ± 3 kg) were randomized into a placebo group (*n* = 12) or an oregano supplementation group (*n* = 12). The participants consumed a capsule containing 500 mg *Origanum vulgare* immediately after completing the ACRT. Blood sampling was taken before exercise, immediately after exercise, and 60 and 120 min after oregano consumption. Plasma levels of creatine kinase (CK), lactate dehydrogenase (LDH), malondialdehyde (MDA), superoxide dismutase (SOD), total antioxidant capacity (TAC), and glutathione peroxidase (GPX) were measured at the four time points. The time × group interactions were found for CK (*p* < 0.0001, d = 3.64), LDH (*p* < 0.0001, d = 1.64), MDA (*p* < 0.0001, d = 9.94), SOD (*p* < 0.0001, d = 1.88), TAC (*p* < 0.0001, d = 5.68) and GPX (*p* < 0.0001, d = 2.38). In all variables, the difference between placebo and oregano groups were significant at 60 (*p* < 0.0001) and 120 (*p* < 0.0001) minutes after ACRT test. The main effect of time was also significant for all the variables (*p* < 0.0001). Our results suggest that oregano supplementation has the potential to reduce muscle damage and increase oxidative capacity following ACRT. Supplementation with oregano may serve as a dietary strategy to increase preparedness and promote recovery in military recruits.

## 1. Introduction

Increasing physical efficiency and preparedness of military forces is a mainstay of military training programs [1,2], where various specialized exercises are used to improve combat preparedness, physical fitness, and mental health [1,3,4]. An important concern associated with intense physical activity is the acute potential damage to the immune and nervous systems and to muscles [5]. High-intensity exercise increases reactive oxidative species (ROS) levels in skeletal muscles, which can reduce force generation [6,7,8]. Extreme levels of physical activity, such as military activity programs and excessive production of ROS [9], decrease antioxidant enzyme levels in the blood and can cause skeletal muscle damage and soreness [10]. Antioxidant enzymes that are upregulated by high-intensity activities [11], such as superoxide dismutase (SOD), catalase (CAT), and glutathione peroxidase (GPX), prevent the deleterious effects of ROS by neutralizing free radicals. Plasma levels of creatine kinase (CK) and lactate dehydrogenase (LDH) are used to monitor markers of muscle damage in response to intense and/or unaccustomed exercise [9,10,12]. For example, CK levels in army recruits increase after 30 km of slow running [13]. A single session of high-intensity resistance exercise increases plasma malondialdehyde (MDA) concentrations (a marker of oxidative stress) [14]. Acute exercise protocols used in army training increase ROS and MDA levels, leading to insufficient recovery, muscle damage, and decreased physical performance [15,16]. Interventions, such as nutritional supplementation during initial military training programs or exercise intensity increases, could provide protection or reduce oxidative stress and exercise-induced muscle damage.

A promising approach to augmenting antioxidant capacity, thereby reducing the effects of ROS and markers of muscle damage, is to consume herbal supplements [17], such as oregano [18,19,20]. Due to its high polyphenol and antioxidant contents, oregano has considerable therapeutic benefits and is among the most important medicinal plants in the world [18,19,20]. The antimicrobial and antioxidant properties of oregano plant extract have been shown [21] and are in part responsible for inhibiting inflammation and inflammatory pathways [21,22].

It has previously been shown that consumption of oregano alone or with rosemary increases antioxidant levels and decreases renal and plasma liver injury markers, such as aspartate transaminase (AST) and alanine transaminase (ALT) in both groups [21]. Moreover, consuming *Origanum vulgare* alone or with rosemary increased antioxidant levels and reduced plasma renal and hepatic damage in both groups [21,22]. The purpose of this study was to investigate the effects of immediate consumption of the oregano plant after the combat readiness test (ACRT) on biomarkers of oxidative stress, antioxidant status, and markers of muscle damage in male army soldiers. Based on the relevant literature [18,19,20,21,22], we hypothesized quicker recovery from exercise in the oregano consumption group compared with the placebo group.

## 2. Materials and Methods

Twenty-four healthy Iranian male soldiers (age: 24 ± 3.5 years, height: 167 ± 14.6 cm, mass: 66 ± 3.2 kg, BMI: 23.7 ± 1.2 kg/m^2^) participated in this study. The G-power program was used to consider the number of subjects. The study was conducted in accordance with the latest version of the Declaration of Helsinki and with the approval of the Research Ethics Committee of Baqiyatallah University of Medical Sciences (IR.SSRC.REC.1396.077). All soldiers were fully informed about the aims and the experimental procedures, and written informed consent was obtained before the start of the study.

None of the participants had a history of using recreational drugs, antioxidants, natural supplements, or alcohol. All participants were members of an identical military garrison, having the same military history, with no history of kidney, liver, cardiovascular disease, diabetes, or other physical injuries. The participants were randomly divided into two experimental groups, oregano treated (*n* = 12) or placebo treated (*n* = 12) (https://www.sealedenvelope.com/simple-randomiser/v1/lists), using a randomized block design with equal block sizes.

### 2.1. Collection and Preparation of Herbal Supplement for Consumption

*Origanum vulgare* was collected from the mountains of Saqez city in the Kurdistan province and dried at room temperature for 10 days. The plant was then dried in an oven for 48 h at 32 °C and powdered using a mortar and pestle. Fifty grams of the sample was used for analysis by gas chromatography–mass spectrometry (GC-MS), and 500 mg was encapsulated for consumption.

### 2.2. Essential Oil Distillation

The sample of *Origanum vulgare* was placed in the oven at a temperature of 32 °C for 24 h to dry. The sample was then powdered with a mortar and pestle. Fifty grams of the powdered sample was extracted for 3 h by aqueous distillation by boiling using a Clevenger distillation apparatus. The extract was filtered and then dried over sodium sulfate without water and finally transferred to a closed glass container and stored at 4 °C. The percent yield of essential oil was calculated as the volume of dried essential oil divided by the mass of the initial dry powder, which yielded a percentage rate of 2.1% [23].

### 2.3. Gas Chromatography–Mass Spectrometry of Oregano

Gas chromatography–mass spectrometry (GC-MS) was used to isolate and identify the volatile components in the essential oregano oil, using an Agilent 5975 mass spectrometer detector (MSD) (Agilent Technologies, Palo Alto, CA, USA) paired with an Agilent USA GC 7890A MS 5975C gas chromatography device (Agilent Technologies, Palo Alto, CA, USA). The column used was HP-5 welded silica (5% phenyl/95% polydimethyl siloxane) with the following specifications: 30 × 0.25 mm^2^ i.d. and 0.25 μm film thickness. Helium was used as a carrier gas, and the mobile phase flow rate was 1 mL/min. The temperature started at 50 °C and was kept constant for 2 min. The temperature program used was as follows: the column temperature began at 60 °C and increased at a rate of 4°/min to 275 °C and remained at this temperature [24]. 

The stored essential oil sample was diluted with n-hexane at a ratio of 1 to 10 and 1 microliter was injected into a gas chromatogram. The injector temperature was 280 °C, and the detector was set at 300 °C. The components of the essential oregano oil were identified by comparing their fragmentation patterns with Wiley7n.l and NIST08 databases and by their retention times in the chromatographic column. The ratios of the sub-peak area to the sum of the sub-peak levels of all compounds were determined for each compound and are summarized Table 1 [24].

### 2.4. Army Combat Readiness Test (ACRT)

To elucidate the effects of oregano consumption on physical fitness, the ACRT was conducted (Figure 1). The soldiers warmed up for 10 min before performing the following activities:(1)400 m running at maximum speed carrying a gun(2)Jumping over low hurdles(3)High crawl(4)Crossing under and over the hurdles(5)40 yards of carrying or dragging a colleague(6)40 yards of carrying ammunition while maintaining one’s balance(7)Targeting while moving(8)100 yards of carrying ammunition while running at a maximum speed(9)100 yards of speed-agility (Figure 1)

### 2.5. Nutrition Control and Consumption of Oregano

The placebo group was given either 500 mg of placebo (starch), and the oregano group was given powder of oregano immediately after the second blood sample. The participants were advised to eat the same specified food two days before the blood sampling (48 h before the start of ACRT) and to not consume anything herbal during this time. The amount of nutrients consumed was calculated using the method described by McCance [25] (Table 2). The participants were also instructed to refrain from consuming coffee, tea, bananas, cereals, and fatty foods at least 24 h prior to testing.

### 2.6. Blood Sampling 

Blood samples were obtained at four time points of the study. Of note, the participants were instructed not to change their diets for at least two days before testing and not to undertake any exercise 48 h prior to testing. A blood sample (10 mL) was obtained from the brachial veins of the participants pre-, immediately post-, 60 min, and 120 min post-ACRT. Blood was drawn into test tubes containing an anticoagulant solution (EDTA) and rapidly centrifuged at 3000 rpm for 10 min. Plasma and serum were separated to determine MDA, TAC, CK, LDH, SOD, and GPx with specific spectrophotometric laboratory kits. Erythrocytes were washed three times with a cold isotonic saline solution, and both erythrocytes and plasma were stored at −80 °C until analysis. 

The plasma concentrations of MDA, TAC, CK, and LDH were determined, and SOD and GPx levels were measured in erythrocytes. The plasma TAC was measured using a chromogenic method with a commercial kit (Cat. No. NX 2332, Randox, Crumlin, UK). The antioxidant capacity of the samples was expressed as millimoles per liter of Trolox equivalents (6-hydroxy-2,5,7,8-tetramethylchroman-2-carboxylic acid). The average intra-assay coefficient of variation (calculated for ten duplicate samples) was 4.9%.

The activities of SOD and GPx were determined in erythrocytes using commercially available kits (RANSOD Cat. No. SD 125 and RANSEL Cat. No. RS 505, respectively; Randox, Crumlin, UK). Antioxidant enzyme activities were measured at 37 °C and expressed in U/g Hb. Hemoglobin was assessed using a standard cyanmethemoglobin method with a diagnostic kit (HG 1539; Randox, Crumlin, UK). The average intra-assay coefficients of variation (calculated for ten duplicated samples) for SOD, GPx, and Hb were 3.8, 6.9, and 2.9%, respectively. Plasma MDA levels were determined with a commercially available kit (LPO-586, OXIS International, Portland, OR, USA). Plasma CK activity was determined with a diagnostic kit (Cat. No. C6512-100, Alpha Diagnostics, San Antonio, TX, USA, USA). The average intra-assay coefficients of variation for MDA and CK (calculated for ten duplicate samples) were 7.5 and 8.3%, respectively.

Concentrations of LDH were measured using a photometric method (DGKC) with a Pars AZ moon quantitative diagnostic kits test (Tehran, Iran) with a coefficient of variation of 1.2% and a sensitivity of 5% international units per liter. Plasma CK levels were measured using the same method with a coefficient of variation of 7% and a sensitivity of 1 international unit per liter.

### 2.7. Statistical Analyses

SPSS version 26.0 (IBM SPSS Statistics) was used in order to analyze data. Statistical significance set at *p* < 0.05. Data values are presented as mean (±SDs). Shapiro–Wilk test was used to check the normal distribution of data. Possible differences between the baseline levels of variables were checked using a *t*-test for independent samples. The effects of the ACRT test on oxidative stress variables were analyzed using a 2 (“group”: oregano vs. placebo) × 4 (“time”: pre vs. post0, post60, and post120) repeated-measures analysis of variance (ANOVA). In the case of statistically significant group × time interactions, the Bonferroni test was used as a post hoc test to identify the statistically significant comparisons. Partial eta squared, taken from ANOVA calculation outputs, was used to estimate effect sizes and then converted to Cohen’s d. Microsoft Excel 2016 was used to design graphs. 

## 3. Results

There were no significant differences between baseline levels of CK in oregano consumption and placebo groups (t = −0.20, *p* = 0.83, 95% CI: −14.3–11.7). The analysis showed a significant group × time interaction (*p* < 0.0001, d = 2.06) for CK. The post hoc analysis revealed a significant difference between the two groups for 60 (*p* < 0.0001, d = 2.84) and 120 (*p* < 0.0001, d = 3.64) minutes in which ACRT significantly increased CK levels in the placebo group compared to the oregano-supplemented group (Figure 2). The main effect of time was significant for CK (*p* < 0.0001, d = 7). Results of the Bonferroni test revealed that all possible paired differences between four times of measurements were statistically significant (*p* < 0.0001) except for the difference between 60 and 120 min after the ACRT test (*p* = 0.06) (Figure 2).

The levels of LDH for placebo and oregano-supplemented groups were not significantly different before the ACRT test (t = 1.37, *p* = 0.18 95% CI: −4.7–23.5). Repeated measures of ANOVA showed a significant group × time interaction (*p* < 0.0001, d = 1.64) for LDH. The results of Bonferroni as a post hoc test showed that there was a significant difference between the two groups for 60 (*p* < 0.0001, d = 3) and 120 (*p* < 0.0001, d = 3.4) minutes, while the differences between the two groups before (*p* = 0.17, d = 0.27) and immediately after (*p* = 0.18, d = 0.29) the ACRT test were not significant (Figure 3). The main effect of time was significant for LDH (*p* < 0.0001, d = 4.89). According to the results of the Bonferroni post hoc test, all differences between the four measurement times were statistically significant (*p* < 0.0001) (Figure 3).

Moreover, the differences between baseline levels of MDA in placebo and oregano-supplemented groups were not significant (t = 1.59, *p* = 0.12, 95% CI: −0.001–0.01). The repeated measures of ANOVA showed a significant group × time interaction (*p* < 0.0001, d = 9.94) for MDA. The results of the Bonferroni test showed that there was a significant difference between the two groups for 60 (*p* < 0.0001, d = 9.93) and 120 (*p* < 0.0001, d = 9.92) minutes, while the differences between the two groups immediately after (*p* = 0.75, d = 0.06) the ACRT test were not significant (Figure 4). The main effect of time was significant for MDA (*p* < 0.0001, d = 9.94). The Bonferroni post hoc test showed that all differences between the four measurement times were statistically significant (*p* < 0.0001) (Figure 4).

There was no significant difference between GPX values of placebo and oregano-supplemented groups (t = 0.96, *p* = 0.34, 95% CI: −1.03–2.83). The groups × time interaction for GPX was significant (*p* < 0.0001, d = 2.38) for GPX. The results of the Bonferroni test showed that there was a significant difference between the two groups for 60 (*p* < 0.0001, d = 3.64) and 120 (*p* < 0.0001, d = 3.84) minutes, while the differences between the two groups immediately after (*p* = 0.16, d = 0.29) the ACRT test were not significant (Figure 5). The main effect of time was significant for GPX (*p* < 0.0001, d = 3.39). The Bonferroni post hoc test showed that all differences between the four measurement times were statistically significant (*p* < 0.0001) (Figure 5).

The differences between the plasma levels of TAC of the placebo and the oregano group at baseline were not significant (t = 0.40, *p* = 0.68, 95% CI: −0.01–0.01). The analysis of data showed that the groups × time interaction for TAC was significant (*p* < 0.0001, d = 5.68) for TAC. The results of the Bonferroni test showed that there was a significant difference between the two groups for 60 (*p* < 0.0001, d = 9.89) and 120 (*p* < 0.0001, d = 4.89) minutes, while the differences between the two groups immediately after (*p* = 0.74, d = 0.07) the ACRT test were not significant (Figure 6). The main effect of time was significant for TAC (*p* < 0.0001, d = 5.68). The Bonferroni post hoc test showed that all differences between the four measurement times were statistically significant (*p* < 0.0001) (Figure 6).

SOD levels were not significantly different at baseline between the placebo and oregano-supplemented groups (t = 0.40, *p* = 0.29, 95% CI: 9.98–0.31–06). Significant groups × time interaction for SOD was reported (*p* < 0.0001, d = 1.88) for SOD. The results of the Bonferroni test showed that there was a significant difference between the two groups for 60 (*p* < 0.0001, d = 3.39) and 120 (*p* < 0.0001, d = 3) minutes, while the differences between the two groups immediately after (*p* = 0.74, d = 0.13) the ACRT test were not significant (Figure 7). The main effect of time was significant for SOD (*p* < 0.0001, d = 4.35). Results of the Bonferroni test revealed that all paired differences between the four different measurement times were statistically significant (*p* < 0.0001) (Figure 7).

## 4. Discussion

Our study examined the effects of oregano consumption on markers of muscle damage and oxidative stress after soldiers’ specialized military drills (ACRT). The intense nature of such a combination (strength, endurance, power, agility) of military exercises predisposes soldiers to muscle injury and oxidative stress, which can challenge immune function [1]. Since soldiers may perform several intense military activities in a day, rapid recovery is of significant importance to this population [1]. There is an increased interest in promoting muscular recovery and antioxidant regulation during intense training using nutritional supplements [17,26,27,28]. Our study’s findings confirm that ACRT increases markers of muscle damage and oxidative stress. They are aligned with previous studies of increased levels of CK, and LDH, which are commonly used markers of muscle damage, after an acute bout of strenuous exercise [26,27,28]. Intense exercise disrupts the sarcolemma, allowing the efflux of intracellular compounds and enzymes such as CK and LDH into the interstitial space and eventually into the circulation [29]. Previous findings have also indicated that oxidative stress causes markers of muscle damage due to ROS production from phagocytic white blood cells [30]. In our study, plasma CK and LDH were lower in the oregano supplement group compared to the placebo group at 60 and 120 min post-ACRT. Our GC-MS data showed that oregano contains carvacrol and thymol, two antioxidant substances that can help maintain the balance of oxidants and antioxidants and reduce markers of muscle damage [31]. While future data are needed to corroborate our findings, the present study suggests that oregano consumption may reduce exercise-induced markers of muscle damage.

Plasma levels of MDA increase after strenuous exercise, likely due to increases in oxygen uptake and free-radical-induced cell membrane degradation, indicating oxidative stress [32]. We showed lower MDA following ACRT in the oregano supplementation group compared to placebo. Antioxidants have an important role in limiting oxidative stress [33]. Mitochondrial SOD and cytoplasmic GPX prevent oxidation and destruction of mitochondrial membranes [34]. Our study demonstrated that supplementation with oregano significantly increases levels of GPX, TAC, and SOD, suggesting greater antioxidant capacity, potentially leading to the lower MDA observed in the oregano supplementation group.

We could not identify other studies using herbal supplements to mitigate the effects of intense exercise on markers of muscle damage in humans. We identified the major constituents of essential oregano oil as cispiritoneapoxide, alpha terpinene, methanone, polygon, piperpronoxide, and ethanol and methanol extract. The high content of phenolic compounds is largely responsible for the high antioxidant activity of some extracts, including methanol and ethanol extracts. There is a positive relationship between the number of phenolic compounds and the antioxidant power of plants [35]. One study found that the essential oil of oregano from Iran, as used in this study, had a free radical inhibitory effect about six times greater than in Turkish oregano [36]. The consumption of *Origanum vulgare* with rosemary increases antioxidant levels and reduces plasma renal and hepatic damage [21]. We could not identify other studies investigating the effects of herbal consumption on muscle damage and oxidative and antioxidant biomarkers induced by ACRT and in army soldiers. It is important to note that some studies have rejected the role of antioxidant supplements in cellular damage based on the use of eccentric exercises, which can damage skeletal muscles less than concentric exercises by causing the secretion of leukocytes [37]. Considering the potential for injury and the risks of the ACRT, it is important to identify the means of preventing or reducing injuries in these individuals. It will be of interest to study the effect of oregano on muscle damage markers and antioxidant enzymes during military physical activity and also military exercises.

### Study Limitations

The aim of this study was to examine the effects of oregano consumption on short-term recovery after military activity, but we did not examine the effects of oregano supplementation on recovery at other time points, such as 24, 48, and 72 h after military activity. Another limitation of this study is the small sample size (only 24 soldiers) and its focus on only military personnel.

## 5. Conclusions

Our study indicates that supplementation with oregano reduces muscle damage and oxidative stress and increases antioxidant markers in male soldiers following intense physical activity of specialized military drills. Supplementation with oregano after ACRT improves MDA, CK, LDH, and total antioxidant capacity. Consuming oregano supplements may serve as a practical means to promote recovery in army soldiers from oxidative stress-induced muscle injury following intense physical activity.

## Figures and Tables

**Figure 1 nutrients-15-00137-f001:**
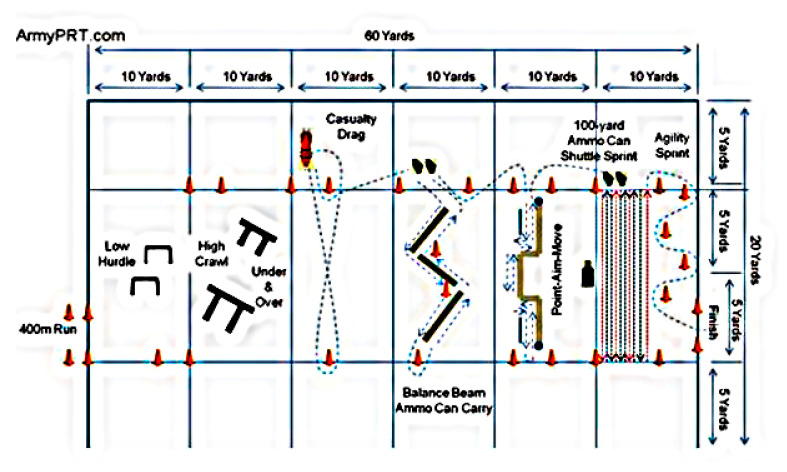
ACRT test steps.

**Figure 2 nutrients-15-00137-f002:**
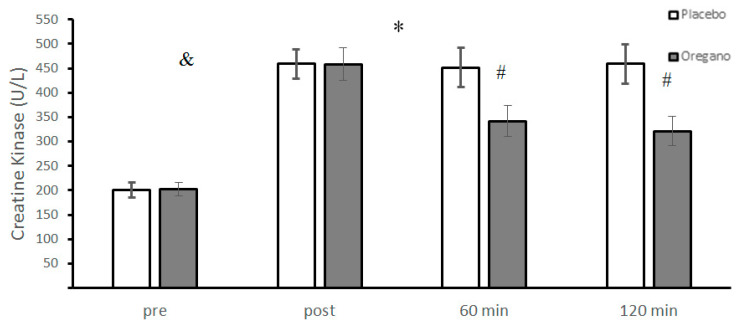
Mean ± standard deviations (SD) of plasma CK levels before and at different times after consuming oregano supplements. The asterisk (*) indicates significant time × group interaction, (#) indicates a significant difference between the two groups, and (&) indicates the main effect of time (*p* < 0.05).

**Figure 3 nutrients-15-00137-f003:**
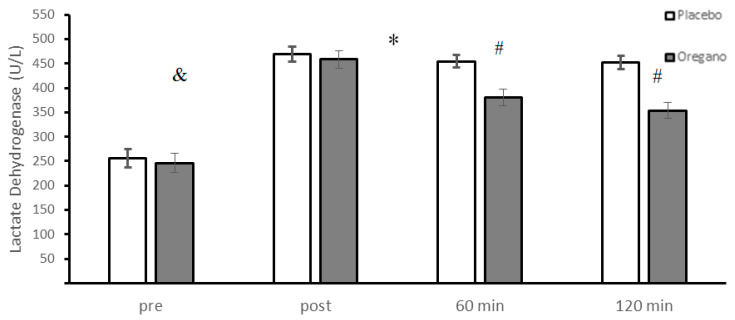
Mean ± standard deviations (SD) of plasma LDH levels before and at different times after consuming oregano supplements. The asterisk (*) indicates significant time × group interaction, (#) indicates significant difference between the two groups, and (&) indicates the main effect of time (*p* < 0.05).

**Figure 4 nutrients-15-00137-f004:**
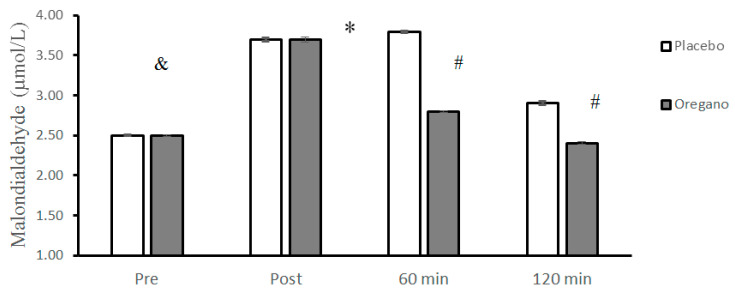
Mean ± standard deviations (SD) of plasma MDA levels before and at different times after consuming oregano supplements. The asterisk (*) indicates significant time × group interaction, (#) indicates a significant difference between the two groups, and (&) indicates the main effect of time (*p* < 0.05).

**Figure 5 nutrients-15-00137-f005:**
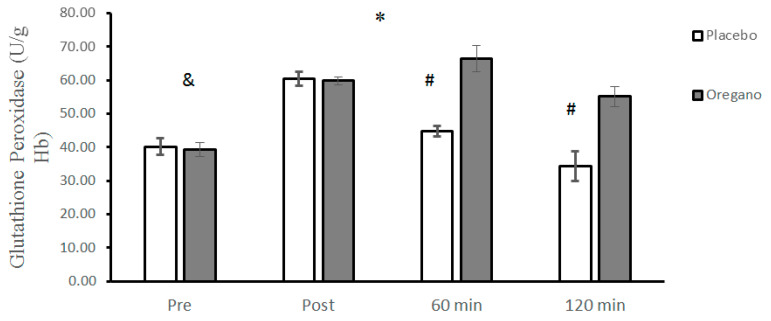
Mean ± standard deviations (SD) of plasma GPX levels before and at different times after consuming oregano supplements. The asterisk (*) indicates significant time × group interaction, (#) indicates a significant difference between the two groups, and (&) indicates significant main effect of time (*p* < 0.05).

**Figure 6 nutrients-15-00137-f006:**
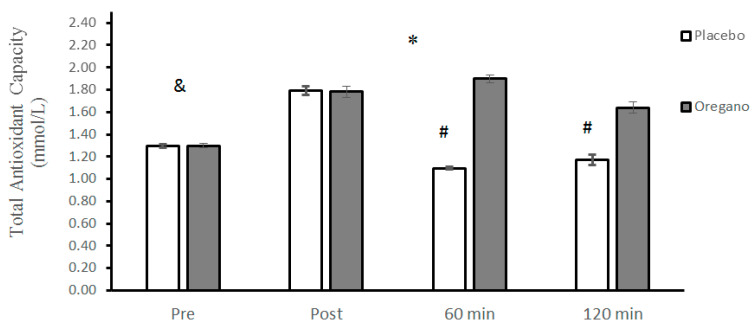
Mean ± standard deviations (SD) of plasma TAC levels before and at different times after consuming oregano supplements. The asterisk (*) indicates significant time × group interaction, (#) indicates a significant difference between the two groups, and (&) indicates the main effect of time (*p* < 0.05).

**Figure 7 nutrients-15-00137-f007:**
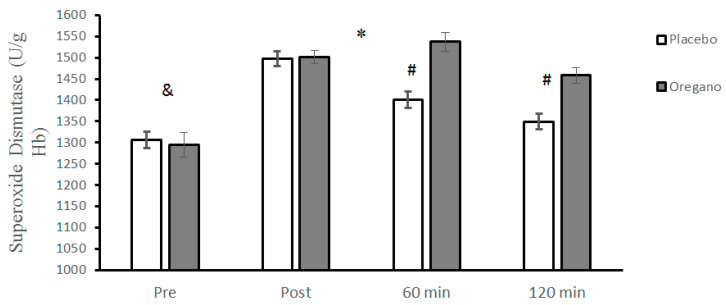
Mean ± standard deviations (SD) of plasma SOD levels before and at different times after consuming oregano supplements. The asterisk (*) indicates significant time × group interaction, (#) indicates a significant difference between the two groups, and (&) indicates the main effect of time (*p* < 0.05).

**Table 1 nutrients-15-00137-t001:** Oregano components.

Component #	Oregano Components	Sub-Peak Area (%)
1	Carvacrol	36.1
2	Thymol	27.8
3	p-Cymene	12.1
4	γ-Terpinene	6.8
5	Methyl carvacrol	4.8
6	β-Pinene	3.2
7	α-Terpinene	1.8
8	Terpinen-4-ol	1.2
9	α-Thujene	1.2
10	Myrcene	0.9

**Table 2 nutrients-15-00137-t002:** Analysis of nutrient supplements (mean ± SD).

	Placebo	Oregano	*p*-Value
Total (kcal/d)	2225 ± 190	2203 ± 157	0.43
Total protein (g)/d	111 ± 16	112 ± 14	0.67
Protein (g/kg BW)/d	1.05 ± 0.45	1.1 ± 0.4	0.58
Total protein (% energy)	18.3 ± 4.1	18.4 ± 4.2	0.86
Total CHO (g)/d	295 ± 19	285 ± 20	0.28
Total CHO (% energy)	48.5 ± 6.9	47.4 ± 7.2	0.39
Total fat (g)/d	81.5 ± 21	83.5 ± 23	0.71
Total fat (% energy)	31.9 ± 7.6	32.5 ± 6.9	0.49

## Data Availability

The data presented in this study are available within the manuscript.

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
