# Peer review of "Oregano (Origanum vulgare) Consumption Reduces Oxidative Stress and Markers of Muscle Damage after Combat Readiness Tests in Soldiers"

_nutrients, 2022, doi:10.3390/nu15010137_

Round 1
Reviewer 1 Report
The objective of the manuscript entitled: “Oregano (Origanum vulgare) consumption reduces oxidative stress and markers of muscle damage after combat readiness tests in soldiers” is the evaluation the effects of oregano supplementation on muscle damage, oxidative stress, and plasma antioxidant markers of soldiers performing the army combat readiness test.
The article is clear, relevant, and presented in a well-structured manner.
There are just a few areas in the article where the results could be expanded to assist the reader in the clinical application of the findings.
Abstract:
1) In what form and amount oregano was supplemented?
Material and methods
2) I suggest describe in details inclusion criteria in the study? This information should be described more widely.
3) „Participants were advised to eat the same specified food two day before the blood sampling”. What was the type of meals? Preparing meals on your own? Diet catering? institutional feeding? etc...?
4) Line 156: Why participants were obliged to exclude cereals and bananas?
5) Line 159: p-value should be added in Table 2
Best wishes
Author Response
Response to Reviewer 1:
|
Reviewer's comments |
Our responses |
|
Abstract:1- In what form and amount oregano was supplemented?
|
Answer: We added this sentence to the Abstract: Participants consumed a capsule containing 500 mg Origanum vulgare immediately after completing the ACRT. |
|
Material and methods 2) I suggest describe in details inclusion criteria in the study? This information should be described more widely.
|
Answer: We added this paragraph in the Material and Methods: None of the participants had a history of using recreational drugs, antioxidants, natural supplements of alcohol. All participants were members of an identical military garrison, having the same military history, with history of kidney, liver, cardiovascular disease, diabetes, or other physical injuries. |
|
3) „Participants were advised to eat the same specified food two day before the blood sampling”. What was the type of meals? Preparing meals on your own? Diet catering? institutional feeding? etc...?
|
Answer: Considering that all the subjects were members of military barracks, they will all consume the same food and in similar quantities. The subjects were advised to strictly follow the barracks food plans during these two days of the study
|
|
4- Line 156: Why participants were obliged to exclude cereals and bananas?
|
Answer: Because cereals and bananas are rich in antioxidants, and consuming carbohydrates and proteins could have direct effects on antioxidant markers and oxidative stress indicators. |
|
5) Line 159: p-value should be added in Table 2
|
Answer: We added p-value in the Table 2. |
Reviewer 2 Report
The present study "Oregano (Origanum vulgare) consumption reduces oxidative stress and markers of muscle damage after combat readiness tests in soldiers" is an interesting topic. However, some issues should be addressed as follows:
1. For Army combat readiness test, how did the authors control the participants' exercise intensity? For example, "400 m running at maximum speed", how did you know it's their maximum speed?
2. Figure 2 should use the high-quality picture.
3. Study limitations should be the subheading of the discussion.
4. Please provide implementation and some future research directions in the discussion part.
5. Please address the more previous relevant research abouth Oregano in the Introduction.
Author Response
Response to reviewer 2:
|
Reviewer's comments |
Our responses |
|
Answer: We used a heart rate monitor to check the intensity of the exercises so that the intensity of the exercise was not less than 90% of the maximum heart rate.
|
|
2. Figure 2 should use the high-quality picture.
|
Answer: Improved Figure 2 |
|
3. Study limitations should be the subheading of the discussion.
|
Answer: Added as suggested |
|
4. Please provide implementation and some future research directions in the discussion part.
|
Answer: We added this paragraph:
It will be of interest to study the effect of oregano on muscle damage markers and antioxidant enzymes during military physical activity and also military exercises .
|
|
5. Please address the more previous relevant research about Oregano in the Introduction.
|
Answer: We added this paragraph in the introduction: Due to its high polyphenol and antioxidant contents, oregano has considerable therapeutic benefits and is among the most important medicinal plants in the world [18-20]. The antimicrobial and antioxidant properties of oregano plant extract have been shown [21], and are in part responsible inhibiting inflammation and inflammatory pathways {Lu, 2008 #1231}[21, 22].
|
Round 2
Reviewer 2 Report
The authors have addressed all questions, and there are no any more questions.
Author Response
The authors have addressed all questions, and there are no any more questions
Answer: we thank the expert reviewer for his/her comments, no further changes are required